# Halogen Bonding Involving Isomeric Isocyanide/Nitrile Groups

**DOI:** 10.3390/ijms241713324

**Published:** 2023-08-28

**Authors:** Andrey S. Smirnov, Eugene A. Katlenok, Alexander S. Mikherdov, Mariya A. Kryukova, Nadezhda A. Bokach, Vadim Yu. Kukushkin

**Affiliations:** 1Institute of Chemistry, Saint Petersburg State University, Universitetskaya Nab. 7/9, 199034 Saint Petersburg, Russia; andrewsmir@yandex.ru (A.S.S.); 195.08pt@gmail.com (E.A.K.); asm93@yandex.ru (A.S.M.); mary_kryukova@mail.ru (M.A.K.); n.bokach@spbu.ru (N.A.B.); 2Laboratory of Crystal Engineering of Functional Materials, South Ural State University, 76, Lenin Av., 454080 Chelyabinsk, Russia

**Keywords:** halogen bond, isocyanides, nitriles, QTAIM, SAPT, NBO analysis

## Abstract

2,3,5,6-Tetramethyl-1,4-diisocyanobenzene (**1**), 1,4-diisocyanobenzene (**2**), and 1,4-dicyanobenzene (**3**) were co-crystallized with 1,3,5-triiodotrifluorobenzene (1,3,5-FIB) to give three cocrystals, **1**·1,3,5-FIB, **2**·2(1,3,5-FIB), and **3**·2(1,3,5-FIB), which were studied by X-ray diffraction. A common feature of the three structures is the presence of I···C_isocyanide_ or I···N_nitrile_ halogen bonds (HaBs), which occurs between an iodine σ-hole and the isocyanide C-(or the nitrile N-) atom. The diisocyanide and dinitrile cocrystals **2**·2(1,3,5-FIB) and **3**·2(1,3,5-FIB) are isostructural, thus providing a basis for accurate comparison of the two types of noncovalent linkages of C≡N/N≡C groups in the composition of structurally similar entities and in one crystal environment. The bonding situation was studied by a set of theoretical methods. Diisocyanides are more nucleophilic than the dinitrile and they exhibit stronger binding to 1,3,5-FIB. In all structures, the HaBs are mostly determined by the electrostatic interactions, but the dispersion and induction components also provide a noticeable contribution and make the HaBs attractive. Charge transfer has a small contribution (<5%) to the HaB and it is higher for the diisocyanide than for the dinitrile systems. At the same time, diisocyanide and dinitrile structures exhibit typical electron-donor and π-acceptor properties in relation to the HaB donor.

## 1. Introduction

Halogen bonding (abbreviated as HaB) is currently among the most actively studied forces in the palette of noncovalent interactions [1,2,3,4,5,6,7,8]. This interest is determined by impressive applications of HaB in various fields of science. Recently published reviews focus on general aspects of HaB [9,10,11,12,13] and also more specific topics such as HaB-based crystal engineering [14,15,16,17,18], sensoring [19,20], molecular [9,20] and anion [11,21] recognition, noncovalent catalysis [11,20,22,23,24,25,26], synthetic organometallic and coordination chemistry [27], polymer chemistry [28], and drug design [29,30,31,32]. It is noteworthy that HaB is involved in human physiology [33,34], in particular in the functioning of thyroid hormones [33].

The great diversity of the studied HaB systems, in many respects, is associated with a significant variety of accepting centers, which act as the nucleophilic components of Hal···Nu contacts. In the overwhelming majority of instances, HaB acceptors are represented by electronegative heteroatoms bearing a lone pair (abbreviated as LP). Owing to the lower electronegativity of C-atoms compared to N, O, S, halogens etc., noncovalent interactions including LP of a carbon site are rare. They have been only identified in some cocrystals of iodoperfluoroarenes with aryl isocyanides [35,36,37] or highly nucleophilic persistent carbenes [38].

In view of our interest in crystal engineering of noncovalent systems with C- or N-nucleophilic coformers, we attempted to compare HaBs, which include the acceptor centers of a different nature in the composition of structurally similar entities and in one crystal environment. For this study, we addressed the isomeric isocyanide and nitrile species and the goal was to establish how the C≡N/N≡C isomerism affects the HaB (Figure 1).

As HaB acceptors, we chose homoditopic 2,3,5,6-tetramethyl-1,4-diisocyanobenzene (**1**), 1,4-diisocyanobenzene (**2**), 1,4-dicyanobenzene (**3**) and, as HaB donor, potentially trifunctional 1,3,5-triiodotrifluorobenzene (1,3,5-FIB) (Figure 2). We found that cocrystallization of diisocyanides **1** and **2** and also dinitrile **3** with 1,3,5-FIB afford cocrystals **1**·1,3,5-FIB, **2**·2(1,3,5-FIB), and **3**·2(1,3,5-FIB), exhibiting halogen-bonded supramolecular architecture. These cocrystals were studied by single-crystal X-ray diffraction (XRD) followed by theoretical calculations to closely interrogate the HaB systems. Our findings—uncovering common features and differences in the two types of noncovalent linkages of isomeric C≡N/N≡C groups (Figure 1)—are detailed in the following sections.

## 2. Results

### 2.1. Crystal Growth and Structural Motifs of the XRD Structures

Diisocyanides **1** and **2** and dinitrile **3** were co-crystallized with 1,3,5-FIB (1:1 molar ratio between the components) on slow evaporation of their solutions at 20–23 °C. This procedure provided three cocrystals, **1**·1,3,5-FIB, **2**·2(1,3,5-FIB), and **3**·2(1,3,5-FIB), which were studied by XRD (Appendix A). The homogeneity of the samples was confirmed by powder diffraction X-ray experiments (Appendix A).

The structures exhibit different molar ratios between the coformers, namely 1:2 for **2**·2(1,3,5-FIB) and **3**·2(1,3,5-FIB), and 1:1 for **1**·1,3,5-FIB; for TG characterization see Appendix A. In all structures, 1,3,5-FIB acts as a trifunctional 120°-orienting HaB donor, whereas the diisocyanides (**1** and **2**) and the dinitrile (**3**) function as homoditopic 180°-orienting HaB acceptors. In the structures of **1**·1,3,5-FIB and **2**·2(1,3,5-FIB), we observed several types of HaBs, namely I···C, σ-(I)-hole···iodine electron-belt, and I···F. In **3**·2(1,3,5-FIB), I···N and σ-(I)-hole···iodine electron-belt contacts. 

Although the crystal structures exhibit different supramolecular architectures, their common feature is the presence of I···C_isocyanide_ or I···N_nitrile_ HaBs occurring between an iodine σ-hole and the isocyanide C- or the nitrile N- atom (Table 1). These HaBs are characterized by rather short interatomic distances. For the diisocyanide cocrystals, the I···N distances are comparable with those previously observed in cocrystals of iodo(perfluoro)arenes with CNMes [35] or with **1** and **2** [37]. For the dinitrile cocrystal, the I···N distance is similar to those found in C_6_F_5_I·NCMes (3.092(4) Å, EBIHEF) [39] and (2,3,5,6-tetramethyl-1,4-dicyanobenzene)·1,4-I_2_C_6_F_4_ (3.061(3) Å, HUMLOQ) [40]. In addition, weak σ-(I)-hole···iodine electron-belt HaB was observed in the structures of **2**·2(1,3,5-FIB) and **3**·2(1,3,5-FIB) and σ-(I)-hole···fluorine, I···F HaB in **1**·1,3,5-FIB (Table 1 and Appendix A).

The diisocyanide and dinitrile cocrystals **2**·2(1,3,5-FIB) and **3**·2(1,3,5-FIB) are isostructural, they exhibit similar cell parameters, and display the same supramolecular organization. In these structures, the I-atoms of the I···C contact (in **2**·2(1,3,5-FIB)) or the I···N contact (in **3**·2(1,3,5-FIB)) function as HaB acceptor sites and are involved in two additional I···I HaBs (Figure 3). In these two cases, two relatively short contacts, which are different in **2**·2(1,3,5-FIB) and **3**·2(1,3,5-FIB), namely I···C and I···N, exist in a very similar environment; this provides a basis for an accurate comparison between the isocyanide and the isomeric nitrile groups. For this comparison, we performed the appropriate quantum chemical calculations detailed in Section 2.2.

In the structure of **1**·1,3,5-FIB, a rather strong HaB is formed, apparently because of an enhanced σ-hole acceptor ability of the isocyano C-atom in **1** due to the combined effect of four electron-donating methyl groups (Figure 4); tri(poly)-center HaBs were not observed.

We measured FTIR-ATR spectra for pure **1**–**3** and their cocrystals (Appendix A). The isocyanide-based cocrystals **1**·1,3,5-FIB and **2**·2(1,3,5-FIB) demonstrate a high-frequency shift of the ν(NC) band (18 and 13 cm^–1^) relative to **1** and **2**, while the difference between the dinitrile-involving structures, **3**·2(1,3,5-FIB) and **3**, is small (1 cm^–1^). This trend in experimental ν(NC) band shifts is in a good agreement with theoretical data (Appendix A). Small changes in ν(NC) frequency for a dinitrile-based system may indicate a weakening of the charge transfer (CT) for dinitrile as compared to isocyanides; this correlates well with the calculation of the CT by NBO method (Section 2.2.4).

The high-frequency shift induced by the HaB resembles the situation with C-isocyanide/N-nitrile coordination to Lewis acidic centers. Siginficant ν(NC) increase (150 cm^–1^) was observed for the H_3_B·CNMe associate and this blue shift was rationalized by an appropriate increase of CN force constant upon coordination [41]. A high-frequency shift of the ν(NC) band in IR and Raman spectra was observed for the homoleptic copper(I) [Cu(CNMe)_4_]^+^ and [Cu(NCMe)_4_]^+^ complexes as compared to the corresponding uncomplexed isocyanide and nitrile species [42]. This shift is greater for the isocyanide ligand (40–60 cm^–1^) and smaller for the nitrile (~15 cm^–1^) [42].

### 2.2. Theoretical Calculations

The nature of C–I···C/N contacts occurring between the diisocyanides or the dinitrile and 1,3,5-FIB were studied by a set of computational methods including Molecular electrostatic potential (MEP), Quantum theory of atoms in molecules (QTAIM), Independent gradient model based on Hirshfeld partition (IGMH), Electron localization function (ELF), Natural Bond Orbital (NBO), The domain based local pair-natural orbital coupled-cluster (DLPNO-CCSD(T)), and Symmetry Adapted Perturbation Theory (SAPT). We also performed a comparative analysis of HaB in the cocrystals to verify common features and differences of these noncovalent interactions.

To study HaB in our systems, geometry optimization was carried out for all bimolecular fragments, [(**1**–**3**)‧1,3,5-FIB]. The optimized structures are shown in Figure 5 and the relevant main geometric parameters are collected in Appendix A. In all structures, the geometry optimization led to a reduction of N≡C (isocyanide) or C≡N (nitrile) and C–I distances (Appendix A), and a slight decrease (by ~6°) of ∠(C–I···C/N). The lengths of the I···C/N HaBs increase by 0.03, 0.08, and 0.10 Å, respectively, for the two diisocyanide structures and one dinitrile structure. In general, the obtained optimized structures are consistent with the experimental XRD data.

#### 2.2.1. Nucleophilicity and Molecular Electrostatic Potential

We determined the global (N_Nu_) and local (N_Nu_^loc^) nucleophilicity indexes and found that N_Nu_ decreases in the following order: 1 (2.35 eV) > 2 (1.80 eV) > 3 (1.40 eV). This order correlates well with Mayr’s experimental nucleophilicity N^+^ indexes [43]. At the same time, the analysis of local nucleophilicity showed that the N_Nu_^loc^ of the C-atom of the isocyanide group in **1** is approximately equal to the N_Nu_^loc^ of the C-atom in **2**. This observation suggests that the interaction energy of HaB should be approximately the same for **1** and **2**. The N_Nu_^loc^ at the N-atom of the nitrile group in **3** is slightly lower than that at any one of the C-atoms of the isocyanide groups (Table 2).

The molecular electrostatic potential (MEP) is also an important descriptor in studies of noncovalent interactions [44,45]. Its analysis reveals the areas of interaction involved in the occurrence of HaB. The calculated MEP isosurfaces for HaB acceptors and their negative potentials (V_s,min_) for HaB acceptors are shown in Figure 6. Its consideration clearly demonstrates the involvement of C- and N-atoms in the HaB. The V_s,min_ value for **3** is slightly larger than those for **2** and **1**; this is in agreement with N_Nu_^loc^ indexes of the HaB acceptors.

#### 2.2.2. QTAIM-IGMH

The QTAIM [46,47] analysis (recommended by IUPAC [48] for studies of HaB) for all contact types revealed the presence of bond critical points (BCPs) and a bond path through the BCP. This bond path indicates the accumulation of the maximum electron density between the interacting nuclear attractors. The most important topological parameters at BCPs (namely, electron density (ρ_b_), Laplacian (∇^2^ρ_b_), and total energy density (H_b_ = V_b_ + G_b_), potential and kinetic energy densities (V_b_ and G_b_), and second eigenvalues of the Hessian matrix (λ_2_) are listed in Table 3 [49]. The molecular graphs for [(**1**–**3**)‧1,3,5-FIB] are given in Figure 7. Low values of ρ_b_ (0.015–0.019 a.u.), positive values of ∇^2^ρ_b_ (0.047–0.050 a.u.), and virtually zero values of H_b_ are typical for noncovalent interactions and their consideration confirms the presence of the closed-shell HaB. The ρ_b_ values for the HaB involving the nitrile are slightly lower than those found for the isocyanides; this indicates a weakening of the interaction energy in the nitrile system. The negative value of λ_2_ ranges from –0.012 to –0.010 a.u., showing that these interactions are attractive. Furthermore, the computed value of the electron localization function (ELF) at the BCP is slightly higher for both (diisocyanide)·1,3,5-FIB systems (Table 3), probably because of a higher contribution of the covalent component in the HaB (for detailed ELF consideration, see Section 2.2.3).

In addition to QTAIM, the HaBs were also explored and visualized using an independent gradient model (IGMH) method [50,51,52,53,54,55]. This method is based on hybrid QM/promolecular atomic separation based on Hirschfeld electron density gradient [56]. The IGMH method also allows for the calculation of IBSI indices, which are useful for the evaluation of HaB strength [57,58].

This IGMH isosurface (Figure 7a,b) is represented by a drop-shaped green surface located between the HaB-donating I- and the HaB-accepting C- or N-atoms. Analysis of the calculated intrinsic bond strength index (IBSI [58], the local stretching force constant) indices, similar to QTAIM, favors the weakening of HaB between 1,3,5-FIB and the dinitrile than that for diisocyanides. In all adducts, no auxiliary interactions were found, which means that HaB is the main attractive interaction in the studied systems.

#### 2.2.3. Electron Localization Function

To understand why the isocyanide systems are characterized by a larger contribution of the covalent component, we additionally performed an ELF analysis [59] for [(**1**–**3**)‧1,3,5-FIB]. This analysis allows us to measure the excess kinetic energy caused by Pauli repulsion and also to visualize the localization of electrons. The values of the ELF function at high electron localization (the localization can be interpreted either as LPs, or chemical bonds) should exhibit a value close to 1, while for low electron localization, the value of the ELF function tends to 0 [60].

The topological analysis of the gradient field of the ELF function is also very useful because this analysis leads to the division of the molecular space into non-overlapping basins of attractors [61]. These basins can be classified as basic (C(X), concentrated on atoms) and valence (V(X) or V(X,Y), concentrated between atoms). The basins could have a synaptic order indicating the number of basins linked together. Figure 8 shows that in all cases, the disynaptic basins between the I-atom of the HaB donor and the C- or N-atoms of the HaB acceptors were not observed; this fact additionally favors the noncovalent nature of the HaB.

Hence, primary attention was paid to the electron population and volume of the monosynaptic V(C) or V(N) and the dinosynaptic V(C,I) basins. The monosynaptic basins can be attributed to LPs on the C- or N- atoms, while the disynaptic basin V(C,I) is related to the localization of the electron density of the C–I bond; these basins are directly involved in the HaBs. The population of the V(C,I) basin for [(**1** and **2**)‧1,3,5-FIB], involved in the HaB is higher (by 0.08 *e*) than the basins that are not involved in the HaB (Table 4).

The population of the monosynaptic basin V(C) is reduced by 0.01 e compared to the unbound C-atoms of the isocyanides. We also found that the interaction of the isocyanides with 1,3,5-FIB leads to a decrease in the local volume of the basin V(C) for LP(C), from 280 to 100 Å^3^ for [**1**‧1,3,5-FIB], and from 440 to 170 Å^3^ for [**2**‧1,3, 5-FIB]. Remarkably, the basin V(C) volume for LP(C) is also decreased from 280 to 100 Å^3^ for [**1**‧1,3,5-FIB] and 440 to 170 Å^3^ for [**2**‧1,3,5-FIB]. The analysis of the ELF basin for [**3**‧1,3,5-FIB] also shows an increase in the population of the V(C,I) basin, but the population of the monosynaptic V(N) basin remains unchanged. All these features are coherent with the weakening of the charge transfer (CT) for the dinitrile system in comparison with the diisocyanide systems.

The ELF data clearly indicate that the covalent component is caused by the outflow of electrons as a result of CT from the diisocyanides (or the dinitrile) to 1,3,5-FIB. However, the HaB with the dinitrile, in contrast to those for both diisocyanides, is characterized by a very weak CT.

#### 2.2.4. Natural Bond Orbital Approach

Since the examination of the ELF results revealed the presence of CT, we performed an NBO analysis to characterize orbital interactions [62] between the HaB donors and acceptors; NBO data are collected in Table 5. We found that two most important donor–acceptor interactions are associated with CT and they are responsible for stabilizing HaB by the HaB acceptors. 

The first type is related to the donor–acceptor interaction with LP(C/N) to σ*-orbital (I–C) of 1,3,5-FIB (Figure 9). The second order perturbation energies E(2) for the transitions LP(C/N)→σ*(I–C) follow the order [**1**‧1,3,5-FIB] (9.6)~[**2**‧1,3,5-FIB] (10.2) > [**3**‧1,3,5-FIB] (5.2 kcal/mol). The second interaction type is associated with the LP(I)⟶σ*/π*(C–N)_isocyanide_ and σ*/π*(N–C)_nitrile_ bond orbitals. On the whole, the E(2) values also follow the general trend of the first type interaction, but are characterized by smaller E(2) values (Table 5). To study the directions of the charge flow, we analyzed the occupancy of the σ*(I–C) orbital of 1,3,5-FIB and also the total population of the orbitals associated with the (C–N)_isocyanide_ or (N–C)_nitrile_ bonds (Table 5). As can be inferred from consideration of the data gathered in Table 5, the population of the σ*(I–C) site increases by 50 me.

This increase is associated with the CT from the LP(C or N) orbitals of the diisocyanides (or the dinitrile) to the σ*(I–C) orbital of 1,3,5-FIB. At the same time, the population of the orbitals of the HaB acceptors and σ*(N≡C) and σ*(C≡N) bonds is increased by 16 me. This increase favors the presence of a reverse CT from LP(I)⟶σ*/π*(C≡N)_isocyanide_ and σ*/π*(N≡C)_nitrile_ bond orbitals. In these cases, the first effect prevails over the second one and it leads to an increase (by 0.01 Å) of the I–C bond length (Appendix A). This observation indicates that for the isocyanides, the direct CT is 3-fold higher than the reverse CT due to the more pronounced σ-donor properties of the diisocyanides compared to the dinitrile. The direct CT decreases in a series [**1**‧1,3,5-FIB] ~ [**2**‧1,3,5-FIB] > [**3**‧1,3,5-FIB]. The latter trend is consistent with the weakening of the nucleophilicity in the order **1**~**2** > **3** (Section 2.2.1; calculated indices are given in Table 2). Thus, the diisocyanides and the dinitrile exhibit typical electron-donor and π-acceptor properties with respect to the HaB donor.

#### 2.2.5. Energy

The interaction (E_int_^SM^) and binding (E_b_^SM^) energy data—obtained using the supramolecular approach and calculated at the DLPNO-CCSD(T)) [63,64] level—are collected in Table 6. In general, E_int_^SM^ values do not exceed –4 kcal/mol; this energy corresponds to weak noncovalent interactions. The values of E_b_^SM^ are comparable to those of E_int_^SM^. This is due to the fact that the deformation energy of the HaB acceptors has a positive sign, while the deformation energy of the HaB donor is negative. Thus, the resultant energy responsible for the deformation of the monomers in the dimer geometry becomes low.

Finally, to closely interrogate the physical nature of the HaBs, we performed the decomposition of the interaction energy by the SAPT [65,66,67,68,69] method into electrostatic (E_elec_), induction (E_ind_), dispersion (E_dis_), and exchange (E_exch_) components (Figure 10). In general, the total interaction energies SAPT(0) (E_int_^SAPT^) correlate with the corresponding energies of the supramolecular interaction (Table 6), although the energies calculated using the DLPNO-CCSD(T) method are 18% lower in absolute values than those calculated at the SAPT(0) level.

According to the obtained SAPT data (Table 6), HaBs are characterized by the predominance of electrostatic energy, which comprise approx. 60% of Σ(E_elec_, E_ind_, and E_dis_). The dispersive attractive energy provides a significant contribution (~30%), while the repulsive Pauli energy has a 70% fraction. It has been proved [70,71] that the high directionality of HaB is determined by electrostatic and exchange-repulsion interactions and therefore we carefully analyzed the E_exch_ energy. Ongoing from [**1**‧1,3,5-FIB] (66%) to [**2**‧1,3,5-FIB] (71%), the contribution of the repulsion Pauli energy increases, and then it decreases in [**3**‧1,3,5-FIB] (43%). This trend is in a good agreement with the weakening of orbital interactions on the transition from the diisocyanide to the dinitrile systems. The highest value of E_exch_ for [**2**‧1,3,5-FIB] is probably associated with an increase in the basin volume LP(C) for [**2**‧1,3,5-FIB] than that for [**1**‧1,3,5-FIB]. As a result of the increase in the LP(C) volume, the E_exch_ energy also increases (Table 4). The inductive effect also makes some contribution to the HaB but it does not exceed 15%. In particular, the induction term includes both polarization and CT. However, the SAPT method allows for an estimate of the contribution of CT to the induction energy [72]. An analysis of the obtained E_ct_ data shows that the contribution of CT to the HaB interaction energy is relatively small (<30% and 6% of the induction energy for the diisocyanides and the dinitrile, respectively).

## 3. Discussion

In this work, we obtained cocrystals of two diisocyanides, **1** and **2**, and one dinitrile, **3**, with HaB-donating 1,3,5-FIB. A characteristic feature of all three structures is the occurrence of C···I or N···I HaB, correspondingly. Two cocrystals, **2**·2(1,3,5-FIB) and **3**·2(1,3,5-FIB), are isostructural, which facilitates the comparison of HaBs with two different acceptors (the terminal C- or N-atoms) under similar crystal environments. The results of our theoretical studies allowed us to identify common features and to verify differences in these two types of interactions (Table 7).

In summary, the examination of the obtained results revealed that the diisocyanides are more nucleophilic than the dinitrile, and they exhibit stronger binding to 1,3,5-FIB. The HaBs in all structures are mostly determined by the electrostatic interactions, which is not, however, sufficient to compensate the repulsion Pauli energy. Considering all these, we led to the conclusion that the dispersion and induction components also provide a noticeable contribution to the HaB and make this interaction attractive. Charge transfer has a small contribution (<5%) to the HaB, but it is higher for the diisocyanide than for the dinitrile systems. At the same time, diisocyanide and dinitrile structures exhibit typical electron-donor and π-acceptor properties in relation to the HaB donor.

The obtained data help enhance the cognition of noncovalent interactions for HaB-involving crystal engineering and provide new perspectives for the targeted design of extended supramolecular architectures utilizing highly directional HaB. The results of this study, in addition, highlight the importance of ditopic 180°-directing HaB acceptors for the construction of linear supramolecular assemblies. Comparison of the isocyanide and nitrile systems shows the significance of using halogen-bonded assembly, which includes C-nucleophilic isocyanide components of noncovalent interactions, for those systems in which strengthening of the HaB due to charge transfer can be expected.

## 4. Materials and Methods

### 4.1. Materials and Instruments

Solvents, compounds **2** and **3**, and 1,3,5-FIB were obtained from commercial sources and used as received. Diisocyanide **1** was prepared from commercial durene according to the known four-step procedure [37]. FTIR-ATR spectra were obtained on Schimadzu IRAffinity-1 instrument (Shimadzu, Kyoto, Japan). The TG studies were performed on a NETZSCH TG 209 F1 Libra thermoanalyzer (NETZSCH, Selb, Germany); MnO_2_ powder was used as a standard. The initial weights of the samples were in the range 1.2–3.9 mg. The experiments were run in an open aluminum crucible in a stream of argon at a heating rate of 10 °/min. The final temperature was 450 °C. Processing of the thermal data was performed with NETZSCH Proteus Thermal Analysis software version 6.1.0 [73]. The samples were examined by XRD phase analysis using Bruker “D8 DISCOVER” high resolution diffractometer (Bruker AXS, Karlsruhe, Germany) with monochromated CuKα long focus source. Background correction and full-profile analyzes of powder diffraction patterns were carried out using the diffractometer software (TOPAS, 4.2).

### 4.2. Cocrystal Growth

1,3,5-FIB (82 mg, 0.16 mmol) and any one of **1**–**3** (0.16 mmol) were mixed in 1:1 molar ratio in hexane/methylene chloride mixture (2 mL; 1:1 v/v), and the obtained solution was left for 2 d at 20–23 °C in a vial with poorly closed lid. During this time, the solvent was completely evaporated and the released crystals of **1**·1,3,5-FIB, **2**·2(1,3,5-FIB), and **3**·2(1,3,5-FIB) were mechanically separated and studied by XRD. 

### 4.3. XRD Studies

XRD experiments were performed using SuperNova (Rigaku Oxford Diffraction, Oxford, UK), Single source at offset/far, HyPix3000 (for **1**·1,3,5-FIB and **2**·2(1,3,5-FIB)) and XtaLAB Synergy (Rigaku Oxford Diffraction, Oxford, UK), Single source at home/near, HyPix (for **3**·2(1,3,5-FIB)) diffractometers with monochromated CuKα radiation. All crystals were kept at 100 K during data collection. The structures were solved using ShelXT [74] structure solution program and refined by means using ShelXL [74] incorporated in Olex2 version 1.5 [75] program package. Empirical absorption correction was accounted for using spherical harmonics implemented in SCALE3 ABSPACK scaling algorithm (CrysAlisPro 1.171.41.122a; Rigaku Oxford Diffraction, 2021). The structures can be obtained free of charge via CCDC database (CCDC numbers 2280076, 2280077, and 2281032).

### 4.4. Computational Details

Full geometry optimization of the model clusters was carried out at the DFT level of theory using the PBE0 [76,77] functional with the atom-pairwise dispersion correction with the Becke–Johnson damping scheme (D3BJ) [78,79]. The ORCA package (version 5.0.3) was used for the calculation [80,81]. Zero-order regular approximation (ZORA) [82] was employed to account the relativistic effects. The optimization calculations based on the X-ray geometries were performed at the PBE0-D3BJ level with the ZORA-def2-TZVP(–f) [82] (for H, C, N, and F) and the SARC-ZORA-TZVP (for I) basis sets [83]. The Hessian matrix was calculated analytically for the optimized structures to prove the location of correct minima (no imaginary frequencies). Combination of the “resolution of identity” and the “chain of spheres exchange” algorithms (RIJCOSX) [84] in conjunction with the auxiliary basis sets SARC/J were used [85]. The SCF calculations were tightly converged “TightSCF”. This level of theory was used the QTAIM, ELF, IGMH, MEP, CDF, NBO, and ETS-NOCV analyses. The natural bond orbital analysis was performed using the NBO 7.0 program [86]. The DLPNO-CCSD(T) method was applied to calculate single-point energy with the ZORA-def2-TZVP(–f) (for H, C, N, and F) and the SARC-ZORA-TZVP (for I) basis. The approach DLPNO-CCSD(T) used the level of accuracy “TightPNO”.

The QTAIM, ELF, IGMH, and MEP calculations were carried out using the Multiwfn 3.8 software [87,88,89] and results were visualized using the VMD program [90]. The SAPT calculations at the SAPT0 level were performed with the recommended basis sets aug-cc-pVTZ (for H, C, N, and F) and aug-cc-pVTZ-PP (for I) for the bimolecular clusters using the Psi4 package (version 1.7.1) [91]. The CDF and ETS-NOCV analyses were carried out according to the methodology described in refs [92,93,94] using the Multiwfn (version 3.8) software.

The interaction and binding energies (E_int_ and E_b_) were calculated for bimolecular clusters as
E_int_(A**···**B) = E(AB) − E{A} − E{B}
E_b_(A**···**B) = E(AB) − E(A) − E(B),
where A and B are molecules, E(AB), E(A), and E(B) are total energies of the corresponding optimized structures, E{A} and E{B} are total energies of A and B in the geometry of the optimized AB.

The global nucleophilicity index was calculated as *N*_Nu_ = E_HOMO_(**1**–**3**) − E_HOMO_(TCE) where E_HOMO_(TCE) energy HOMO tetracyanoethylene (TCE) molecule [95]. The local nucleophilicity index *N*_Nu_^loc^ where calculated as *N*_Nu_^loc^ = f_x_ − *N*_Nu_ were f_x_^−^ is Fukui function of atom x in a molecule [96].

## Figures and Tables

**Figure 1 ijms-24-13324-f001:**
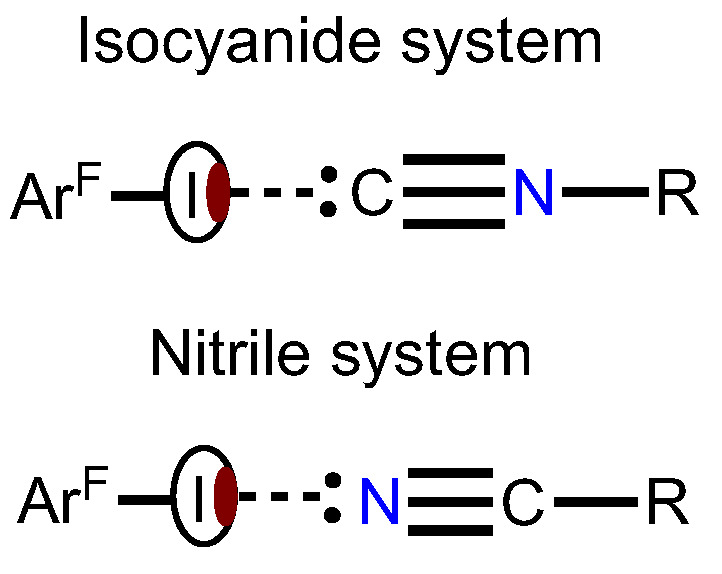
Schematic representation of HaB in the isomeric isocyanide and nitrile systems.

**Figure 2 ijms-24-13324-f002:**
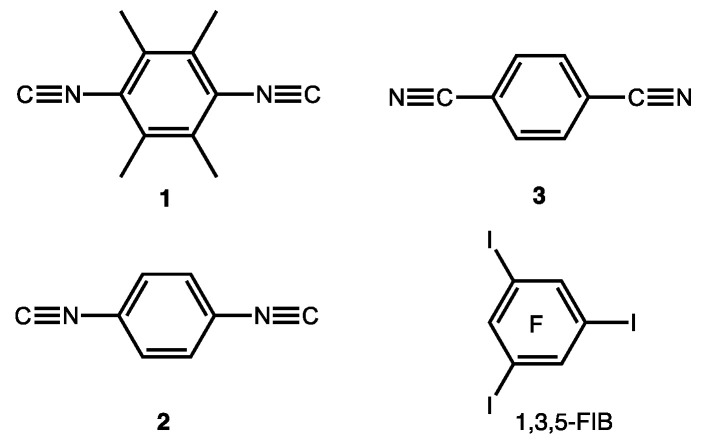
Coformers of the HaB-based cocrystals and their numbering.

**Figure 3 ijms-24-13324-f003:**
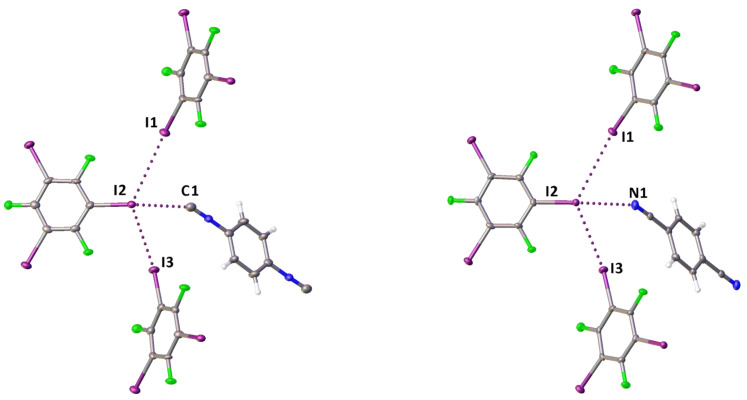
Fragments of the molecular structures of **2**·2(1,3,5-FIB) (**left**) and **3**·2(1,3,5-FIB) (**right**); HaB is given by dotted lines.

**Figure 4 ijms-24-13324-f004:**
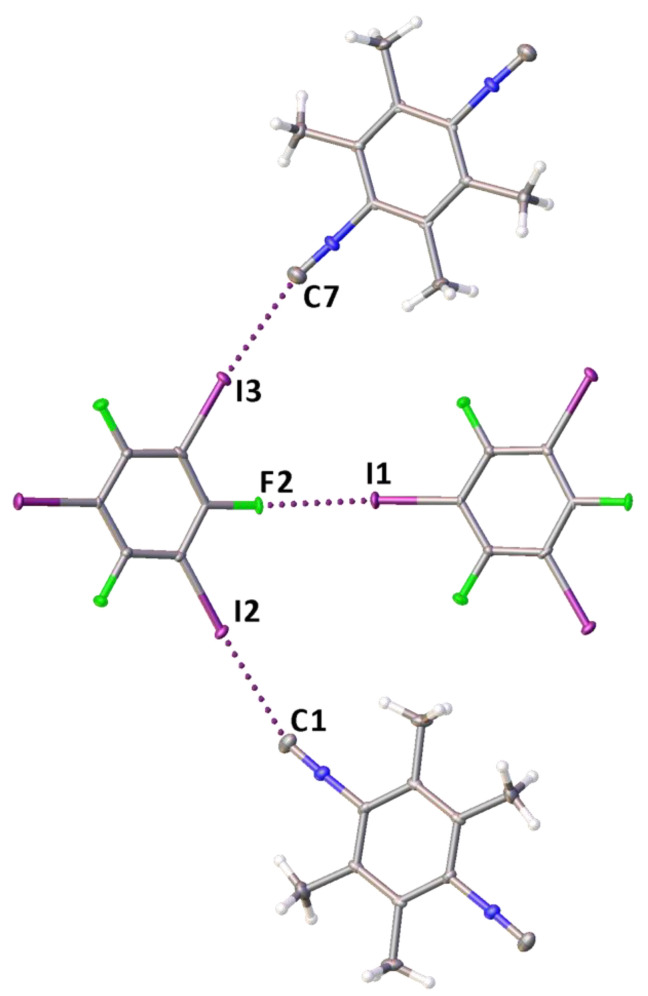
A fragment of the molecular structure of **1**·1,3,5-FIB displaying three two-center HaBs.

**Figure 5 ijms-24-13324-f005:**
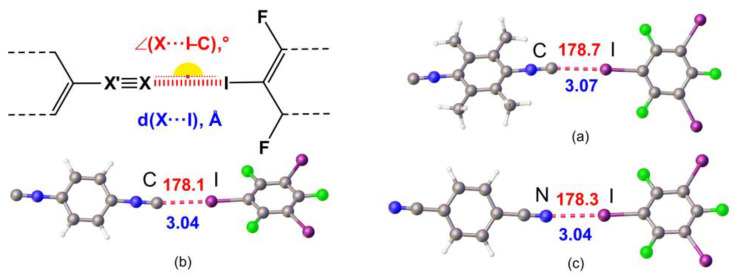
Optimized structures of (**a**) [**1**‧1,3,5-FIB], (**b**) [**2**‧1,3,5-FIB], and (**c**) [**3**‧1,3,5-FIB].

**Figure 6 ijms-24-13324-f006:**
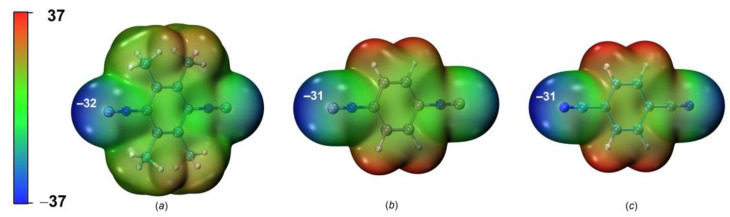
MEP distribution in (**a**) **1**, (**b**) **2**, and (**c**) **3** calculated for the optimized structures (*V_s_*_,min_ MEP values in kcal/mol).

**Figure 7 ijms-24-13324-f007:**
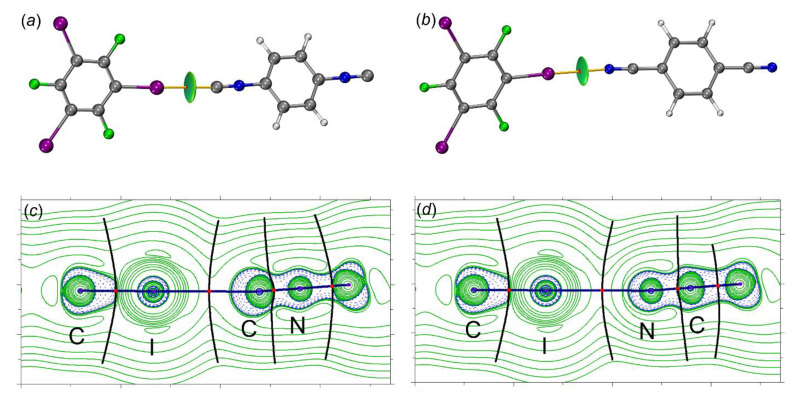
The molecular graph (BCPs in red and bond paths as orange lines) and δg_inter_ isosurface for the structure of (**a**) [**2**‧1,3,5-FIB] and (**b**) [**3**‧1,3,5-FIB] (δg_inter_ = 0.01 a.u. and blue-cyan-green-yellow-red color scale −0.05 < sign(λ_2_)ρ(r) < 0.05); Contour line diagram of the Laplacian distribution, ∇2ρ(r), zero flux surfaces and bond paths in the C–I–C and C–I–N plane (bond critical points are shown by red dots) for the systems of [**2**‧1,3,5-FIB] (**c**) and [**3**‧1,3,5-FIB] (**d**).

**Figure 8 ijms-24-13324-f008:**
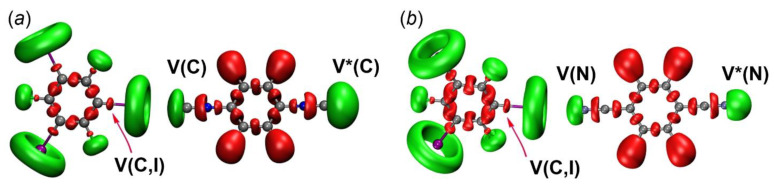
ELF localization domains for the structures of (**a**) [**2**‧1,3,5-FIB] and (**b**) [**3**‧1,3,5-FIB]. Monosynaptic basins are shown in green, disynaptic basins are shown in red. * Values of the basin V(C or N) that are not involved in the HaB.

**Figure 9 ijms-24-13324-f009:**
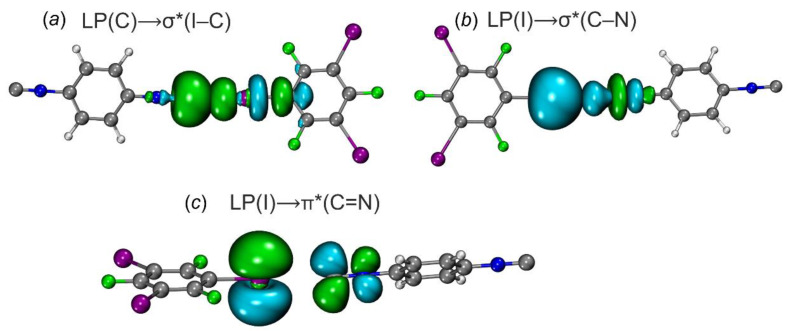
Natural bond orbitals corresponding to the (**a**) LP(C)→σ*(I−C) transitions; (**b**) LP(I)→σ*(C−N); (**c**) LP(I)→π*(C=N) in [**2**‧1,3,5-FIB]. Green and blue colors indicate the opposite phase of the wave function.

**Figure 10 ijms-24-13324-f010:**
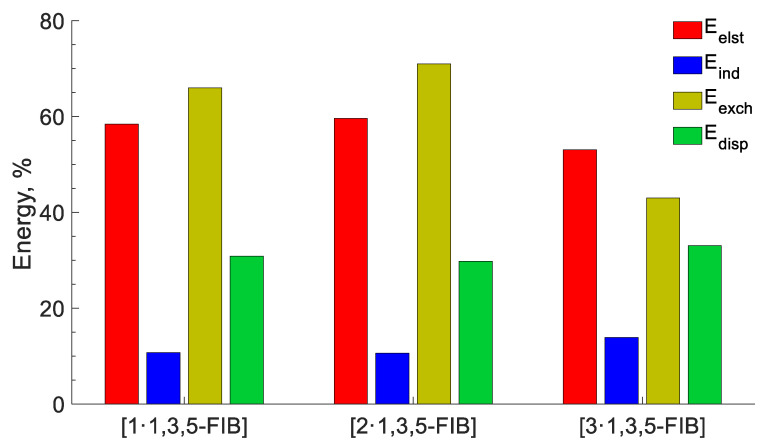
Contribution to the sum of the attractive contributions to the interaction energy (%). Energies are in kcal/mol.

**Table 1 ijms-24-13324-t001:** Geometrical parameters of HaB.

Cocrystal	Contact I···Nu (Nu = C, I, or F)	Interatomic Distance, Å	Nc ^1^	Angle C–I···Nu, °	Angle I···Nu–Y, °
**1**·1,3,5-FIB	I2···C1	3.084 (5)	0.84	173.22 (16)	153.4 (4)
	I3···C7	3.035 (5)	0.82	173.38 (16)	159.5 (4)
	I1···F2	2.992 (3)	0.87	171.54 (13)	136.9 (3)
**2**·2(1,3,5-FIB)	I2···C1	2.957 (6)	0.80	172.0 (2)	138.4 (5)
	I1···I2	3.9561 (5)	1.00	163.51 (14)	116.60 (15)
	I3···I2	3.8022 (4)	0.96	172.52 (14)	110.56 (14)
**3**·2(1,3,5-FIB)	I2···N1	2.935 (5)	0.80	173.79 (19)	135.9 (5)
	I1···I2	3.9625 (6)	1.00	163.79 (14)	116.95 (18)
	I3···I2	3.8122 (5)	0.96	171.32 (14)	111.63 (17)

^1^ Nc is the normalized contact, which here is the ratio between the observed I···C, I···N, I···F, and I···I distances and the sum of the van der Waals radii of involved atoms.

**Table 2 ijms-24-13324-t002:** Global and local nucleophilicity of HaB acceptors.

	1	2	3
*N*_Nu_, *e*V	2.35	1.80	1.40
*N*_Nu_^loc^ (C or N), *e***e*V	0.33	0.30	0.23

**Table 3 ijms-24-13324-t003:** Electron density (ρ_b_), its Laplacian (∇^2^ρ_b_), potential, kinetic, and total energy densities (V_b_, G_b_, and H_b_), second eigenvalue of the Hessian matrix (λ_2_), values (in a.u.), ELF at BCPs, IBSI, and δg^pair^.

Contact	Clusters	ρ_b_	∇^2^ρ_b_	V_b_	G_b_	H_b_	λ_2_	ELF	IBSI	δg^pair^
I···C	[**1**‧1,3,5-FIB]	0.0177	0.0467	–0.0101	0.0109	0.0008	–0.0115	0.09	0.032	0.027
I···C	[**2**‧1,3,5-FIB]	0.0186	0.0494	–0.0109	0.0116	0.0007	–0.0123	0.09	0.034	0.028
I···N	[**3**‧1,3,5-FIB]	0.0150	0.0505	–0.0094	0.0110	0.0016	–0.0103	0.05	0.025	0.024

**Table 4 ijms-24-13324-t004:** Values of the basin electron population V(C or N), V(C,I), (in *e*), and electron volume V_ELF_ for basin V(C or N) (in Å^3^).

Clusters	V(C or N), *e* [V_ELF_, Ǻ^3^]	V(C,I), *e*
[**1**‧1,3,5-FIB]	2.62 [100]; 2.63 * [380]	1.75; 1.67 *; 1.67 *
[**2**‧1,3,5-FIB]	2.60 [170]; 2.61 * [610]	1.76, 1.68 *; 1.68 *
[**3**‧1,3,5-FIB]	3.31 [90]; 3.31 * [460]	1.74, 1.67 *; 1.67 *

* Values of the basin V(C or N) and V(C,I) electron population that are not involved in the HaB.

**Table 5 ijms-24-13324-t005:** Second order NBO perturbation energies (E(2), in kcal/mol), change of the occupancy of the σ*(I–C) and σ*/π*(C≡N) or σ*/π*(N≡C) NBO on the occurrence of HaB (Δocc, in me).

Clusters	Transition	E(2)	Δocc
[**1**‧1,3,5-FIB]	LP(C)⟶σ*(I–C)	9.6	44
	LP(I)⟶σ*/π*(C≡N)	2.1	15
[**2**‧1,3,5-FIB]	LP(C)⟶σ*(I–C)	10.2	46
	LP(I)⟶σ*/π*(C≡N)	2.3	16
[**3**‧1,3,5-FIB]	LP(N)⟶σ*(I–C)	5.2	16
	LP(I)⟶σ*/π*(N≡C)	1.8	12

**Table 6 ijms-24-13324-t006:** Calculated values of the interaction and binding energies and their decomposition (in kcal/mol).

Cluster	E_elst_	E_ind_ (E_ct_)	E_exch_	E_disp_	E_int_^SAPT^ *	E_int_^SM^	E_b_^SM^
[**1**‧1,3,5-FIB]	–8.7	–1.6 (0.5)	9.8	–4.6	–5.1	–4.2	–4.1
[**2**‧1,3,5-FIB]	–9.0	–1.6 (0.5)	10.6	–4.5	–4.6	–3.8	–3.8
[**3**‧1,3,5-FIB]	–6.1	–1.6 (0.1)	6.4	–3.8	–5.0	–3.7	–3.7

* E_int_ (SAPT) = E_elst_ + E_ind_ + E_dis_ + E_exch_ + δ_HF_.

**Table 7 ijms-24-13324-t007:** Comparison of the diisocyanides and dinitriles functioning as HaB acceptors.

Methods (Descriptors)	HaB I···C^iso^ vs. I···N^nitr^	Comments
Global Nucleophilicity (N_Nu_)	N_N_^iso^ > N_N_^nitr^	Diisocyanides are more nucleophilic than the dinitrile
Local Nucleophilicity (N_Nu_^loc^)	N_Nu_^loc^(iso) > N_Nu_^loc^(nitr)	
MEP (V_s,min_)	V_s,min_^iso^~V_s,min_^nitr^	Electrostatic potentials are nearly the same for both systems
QTAIM (ρ_b_)	ρ_b_^iso^ > ρ_b_^nitr^	The electron density values at the BCP of the HaB are slightly higher for the diisocyanide system; this indicates a stronger binding of the diisocyanides to 1,3,5-FIB
IBSI	IBSI^iso^ > IBSI^nitr^	The index IBSI indicates a stronger binding of the diisocyanides to 1,3,5-FIB
ELF (V(C,I))	V(C,I)^iso^ > V(C,I)^nitr^	Charge transfer effect is higher for the isocyanide systems
NBO (Δocc (σ*(I–C))	Δocc ^iso^ > Δocc ^nitr^	
DLPNO-CCSD(T) (E_b_^SM^)	E_b_^iso^ > E_b_^nitr^	Binding energy between the coformers is larger for the diisocyanide structures
SAPT (E_exch_)	E_exch_^iso^ > E_exch_^nitr^	Repulsive Pauli energy for the I···C/N HaBs is higher for the isocyanides. This is in agreement with increased orbital interactions for the diisocyanide systems
SAPT (E_elst_)	ΔE_elst_^iso^~ΔE_elst_^nitrl^	SAPT results indicate the dominance of Coulomb interactions in the HaB of both systems; they contribute approximately 60% to the sum of negative energy components. The dispersion contribution is no more than 30%
SAPT (E_disp_)	ΔE_disp_^iso^~ΔE_disp_^nitrl^	

## Data Availability

The data presented in this study are available in the article and Appendix A. Also CIFs are openly available in www.ccdc.cam.ac.uk/data_request/cif (accessed on 23 May 2023; version 5.44).

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
