# Peer review of "Halogen Bonding Involving Isomeric Isocyanide/Nitrile Groups"

_ijms, 2023, doi:10.3390/ijms241713324_

Round 1
Reviewer 1 Report
This work by Kukushkin and co-workers presents a comparison of the two types of non-covalent linkages of C-N/N-C groups in structurally similar entities and in one crystal environment. They investigate the bonding environments with a set of theoretical methods. The work appears to have been very carefully undertaken. The impact of this work, to a wider audience, could be greatly increased by adding couple of sentences in the conclusions stating the significance of their findings.
Author Response
We thank the reviewer for his/her positive assessment of our study.
According to the reviewer’s recommendations we added the following paragraph in the conclusions section: “The obtained data help enhance the cognition of noncovalent interactions for HaB-involving crystal engineering and provide new perspectives for the targeted design of extended supramolecular architectures utilizing highly directional HaB. The results of this study, in addition, highlight the importance of ditopic 180°-directing HaB acceptors for the construction of linear supramolecular assemblies. Comparison of the isocyanide and nitrile systems shows the significance of using halogen-bonded assembly, which includs C-nucleophilic isocyanide components of noncovalent interactions, for those systems in which strengthening of the HaB due to charge transfer can be expected. ”
Reviewer 2 Report
Paper " Halogen Bonding Involving Isomeric Isocyanide/Nitrile Groups" by A.S. Smirnov et al compares HaBs formed by the acceptor centers of different nature in the composition with structurally similar entities and in the same crystal environment. The isomeric isocyanide and nitrile species are considered to establish how the CN/NC isomerism affects the HaB. Two diisocyanides and one dinitrile were co-crystallized with 1,3,5-triiodotrifluorobenzene (1,3,5-FIB) and these cocrystals were studied by both XRD single-crystal analysis and theoretical calculations. The common features and differences in the two types of noncovalent linkages of isomeric CN/NC groups were studied.
The results of this study show that the diisocyanides are more nucleophilic than the dinitrile and their stronger binding to 1,3,5-FIB takes place. The HaBs in all structures are electrostatically driven, however electrostatics does not compensate the repulsive Pauli energy. Therefore, the dispersion and induction contributions make this HaB interaction attractive. The diisocyanide and dinitrile demonstrate typical electron-donor and pi-acceptor properties in relation to the HaB donor.
Comments.
1) The monochromated CuKα radiation was used in XRD analysis. That means the X-ray reflections within the reciprocal space sphere of R < 0.7 Angstr**-1 were collected. It yields too little resolution, which leads to strong distortions of I-X bond lengths. It also will influence the other bond lengths in table 1.
2) This work exploits descriptors of different nature. These are Molecular electrostatic potential (MEP), Quantum Theory of Atoms in Molecules (QTAIM), Independent Gradient Model (IGMH), Electron localization function (ELF), Natural Bond Orbital (NBO), as well as the Symmetry Adapted Perturbation Theory (SAPT). These descriptors carry different information on the bonding situation. I think that comprehensive explanation of content of
all descriptors should be neccessary done.
3) Application of Independent gradient model (IGMH) is unneccessary in this paper.
Author Response
1) The monochromated CuKα radiation was used in XRD analysis. That means the X-ray reflections within the reciprocal space sphere of R < 0.7 Angstr**-1 were collected. It yields too little resolution, which leads to strong distortions of I-X bond lengths. It also will influence the other bond lengths in table 1.
We agree with the reviewer that, for example, Mo radiation would be better for structures bearing heavy atoms (iodine, in this particular case). However, rather small crystals weakly diffract with the Mo radiation in view of the limited technical characteristics of the available SuperNova (Mo) diffractometer, therefore we choose Cu-Kα, whereupon all data were tested by checkcif which did not revealed mistakes. We added the following phrase in Experimental Section: “Empirical absorption correction was accounted for using spherical harmonics implemented in SCALE3 ABSPACK scaling algorithm (CrysAlisPro 1.171.41.122a; Rigaku Oxford Diffraction, 2021).”
2) This work exploits descriptors of different nature. These are Molecular electrostatic potential (MEP), Quantum Theory of Atoms in Molecules (QTAIM), Independent Gradient Model (IGMH), Electron localization function (ELF), Natural Bond Orbital (NBO), as well as the Symmetry Adapted Perturbation Theory (SAPT). These descriptors carry different information on the bonding situation. I think that comprehensive explanation of content of all descriptors should be neccessary done.
The original submission contains a brief description and appropriate references to the theoretical methods used to study noncovalent interactions in the reported systems. This summary and references, we think, should allow the reader to become familiar with each method. A more detailed description of the standard theoretical methods, in our opinion, should make the article overloaded.
However, following the reviewer recommendations, we updated short descriptions and added additional references to sections 2.2.1. Nucleophilicity and molecular electrostatic potential, 2.2.2. QTAIM-IGMH, and 2.2.4. Natural Bond Orbital approach.
3) Application of Independent gradient model (IGMH) is unneccessary in this paper.
NCI analysis, or IGMH as its version, is an important addition to the QTAIM method in studies of noncovalent interactions (for details see doi: 10.1016/B978-0-12-821978-2.00076-3). Remarkably, the IGMH method is more reliable than the NCI method and we successfully used the former approach for investigations of various types of noncovalent interactions (see, e.g., 10.1021/acs.cgd.0c01474, 10.1039/d3qi00087g, 10.1021/jacs.1c06498). Moreover, IGMH allows not only to visualize, but also to provide a quantitative assessment of the strength of the interaction using the IBSI indices. Therefore, in this report, we used both methods since they complement each other and provide a more detailed picture of noncovalent interactions. We hope our explanation is satisfactory and we suggest leaving the section, which is focused on IGMH, as is.
Round 2
Reviewer 2 Report
Authors revision increases the soundness of the MS strongly. However, their opinion on the Independent gradient model (IGMH) is wrong. Despite of the numerous applications of this descriptor in a literature, the IGMH method says nothing about the atomic interactions. The fact of matter is that IGMH does not carry any information on chemical bonding because of the SPHERICAL non-interacting atoms are used in the model. that is bonding dissappears For noncovalent interactions, IGMH shows the picture, which resembles the true one. However, this resemblance is just imagery. Therefore, by construction, IGMH, does not allows visualizing
and providing a quantitative assessment of the atomic interactions.
Author Response
All drawbacks, indicated by the reviewer, are related to the IGM method, whereas in our work we used the other approach, namely the IGMH (Independent gradient model based on Hirshfeld partition) method, which is free of all these drawbacks and is perfectly useful to characterize halogen bonds in the systems under study. The IGMH method differs from the original IGM in two main aspects:
- The free-state atomic density ρfreei in the IGM formulae is replaced with ρHirsh. It is noted that Hirshfeld partition is a commonly employed method with a clear physical picture to obtain atomic densities in a chemical system.
- The sign(λ2)ρ is evaluated based on actual density similar to the NCI and IRI methods.
Since the IGMH is defined fully based on actual density and gets rid of promolecular approximation, it can be expected that IGMH should have certain practical benefits in exhibiting interactions compared to the IGM approach. All our arguments are based on the original article by Dr. Tian Lu (Beijing Kein Research Center for Natural Sciences) [10.1002/jcc.26812]. In our work, we did not find any limitations for our simple system, therefore we consider it appropriate to use this method for visualizing noncovalent interactions and we suggest leaving this section as is. In principle, if the Editor wish so, this manuscript may be sent to Dr. Tian Lu as a judicator.
In the revision-II, we fully deabbreviated IGMH to avoid further confusions.

Round 3
Reviewer 2 Report
The authors refer to the authority of Dr. Liu. This is not a scientific argument. The approach they use, as its name suggests, is based on the promolecule approximation to isolate atomic contributions. According to Hirschfeld, these atomic contributions are proportional to the contributions of electron density of each spherical atom to the promolecule. This is a serious assumption, and it's bad that the authors don't want to admit it. This should at least be indicated in the text of the article, so that there is no question whether the authors understand the meaning of the descriptors they use.
Author Response
The authors refer to the authority of Dr. Liu. This is not a scientific argument. The approach they use, as its name suggests, is based on the promolecule approximation to isolate atomic contributions. According to Hirschfeld, these atomic contributions are proportional to the contributions of electron density of each spherical atom to the promolecule. This is a serious assumption, and it's bad that the authors don't want to admit it. This should at least be indicated in the text of the article, so that there is no question whether the authors understand the meaning of the descriptors they use.
We thank the reviewer for his/her valuable comments on the IGMH method. In general, each theoretical method is based on its own approximation and, consequently, has its own advantages and disadvantages. This also applies to IGMH, which we used to visualize noncovalent interaction, proved by other methods, namely by QTAIM, ELF, and NBO. Since we are only users of already developed theoretical methods, we provided just a minimal description of the IGMH method and added links to relevant sources, as suggested by the reviewer: “This method is based on hybrid QM/promolecular atomic separation based on Hirschfeld electron density gradient. [10.1002/jcc.27123]”. Our work is far from discussion of the advantages and disadvantages of theoretical methods and, we feel, a detailed discussion of the IGMH method is out of the scope of our manuscript.
Round 4
Reviewer 2 Report
I think that improved manuscript may be published in IJMS.
General check of this manusript is needed.